# A qualitative study of abortion professionals' perspectives on doula support

Martha Paynter[1]*, Anja McLeod[2], Clare Heggie[3], Wendy V. Norman[4]

1 Faculty of Nursing,University of New Brunswick, Fredericton, Canada, 2 Department of Sociology, Dalhousie University, Halifax, Canada, 3 Department of Interdisciplinary Studies, University of New Brunswick, Fredericton, Canada, 4 Faculty of Medicine, University of British Columbia, Vancouver, Canada

* martha.paynter@unb.ca

## Abstract

### Objective(s)

Doulas provide non-clinical support to people in pregnancy, labour and delivery, and, increasingly, to abortion-seekers. Even in contexts of decriminalization and proactive policy efforts to enhance access, barriers to abortion persist, especially for patients experiencing intersecting forms of oppression. Abortion doulas may play a role in improving access. This qualitative study explores the potential involvement of doulas in abortion services from the perspective of abortion health professionals and clinic administrative staff.

### Study design

This is a qualitative study. Participants were asked about experiences with doulas and potential doula contributions to abortion services. Data were analysed using thematic analysis.

### Results

We conducted two focus groups and five individual interviews for a total of 18 participants. Participants perceived challenges to incorporating doulas formally into the abortion clinic setting, although they saw the possibility for doula support before and after receipt of abortion services.

### Conclusion(s)

Our findings suggest abortion health professionals and clinic administrative staff perceive roles for doulas largely external to the clinical setting, to provide support for equity-deserving groups. They have concerns about the presence of doulas in the clinic due to policies against support people and because of the unregulated, unstandardized nature of doula practice.

**Data availability statement:** Data cannot be shared publicly due to ethical obligations to protect the anonymity of participants. Requests for data can be sent to ethics@unb.ca for researchers who meet the criteria for access to confidential data.

**Funding:** MP and WVN received funding for this project from Health Canada's Sexual and Reproductive Health Fund under the Health Care Policy and Strategies Program, as part of University of British Columbia's "CART Access project: Advancing access to abortion for under-served populations through tools for healthcare professionals and people seeking care". Grant number: 2223-HQ-000258. Funders did not play any role in the study design, data collection and analysis, decision to publish, or preparation of the manuscript. https://www.canada.ca/en/health-canada/services/funding/sexual-reproductive-health-fund.html WVN was supported as an Applied Public Health Research Chair by the Canadian Institutes of Health Research and Public Health Agency of Canada (2014–2024, CPP-329455) and as a Tier 1 Canada Research Chair in Family Planning Innovation (2024–2032, CRC-2023-00135).

**Competing interests:** The authors have declared that no competing interests exist.

## Implications

This qualitative study addresses the potential for integration of doulas with abortion care from the perspective of abortion clinic health professional and administrative staff. To our knowledge, this is the first study on this topic in Canada.

## Introduction

### Background

Doulas provide non-clinical support to people in pregnancy, labour and delivery, and, increasingly, to abortion-seekers. Even in contexts of decriminalization and proactive policy efforts to enhance access, barriers to abortion may persist, including social stigma, lack of health professional training, limited gestational duration clinical capabilities, costs such as childcare and transportation, and direct fees for people without public or private health insurance. [1–5] Patients experiencing intersecting forms of oppression, including homophobia and transphobia, racism, violence, poverty and ableism, may face additional barriers. Abortion doulas may play a role in improving access. This qualitative study sought to explore the potential involvement of doulas in abortion services from the perspective of abortion health professionals and clinic administrative staff.

### Abortion doulas

A doula is a non-clinical support person who provides emotional, physical, and informational support throughout the perinatal period [6]. The doula profession is unregulated although there are several training and certifying organizations. Doula support in the birth and postpartum period is associated with reductions in peripartum depression, fewer cesarean deliveries, increases in breastfeeding initiation, and improved patient experience [7–12]. Birthing people facing racism, poverty, and homophobia/transphobia may benefit from the support of a doula during labour [13–15]. In an effort to improve outcomes for equity-deserving people – defined as those who face barriers to accessing equal resources, opportunities, and outcomes because of systemic discrimination [16] – there is mounting advocacy for formalized partnerships between doula organizations and clinical institutions [13,17–21].

Increasingly, doula support is available for all pregnancy outcomes, including abortion, miscarriage, stillbirth, infant loss, assisted reproduction, surrogacy, child protection involvement, and adoption. Existing research about abortion doulas has found clients report positive experiences [22–26].

Our research team conducted an international scoping review of the role of doulas and community birth workers in abortion and contraception care [24]. We found doula roles include information provision, service navigation, physical and emotional support, and in jurisdictions with restrictive abortion policies, direct care. Abortion doulas improved client satisfaction and mitigated barriers to accessing abortion [24]. These studies are largely US-based and focused on first-trimester procedural abortion,

and/or doula support for pain. We identified no studies in Canada. Chor et al. [27] explored clinic staff and abortion doula perspectives on the integration of doula support in a high-volume urban procedural abortion clinic. Participating physicians reported initial challenges including consistency, communication, and the busy clinical setting, but once implemented, all clinic staff endorsed the program.

We conducted an environmental scan of the doula workforce in Canada, identifying 699 doulas with publicly listed services [28]. Of these, 10% identified themselves as supporting abortion or termination of pregnancy, and 5% used the term "full-spectrum"; a term frequently used by doulas who support all pregnancy outcomes [29]. In 2023, we conducted a qualitative study with abortion doulas across Canada [30]. Participants described themselves as system navigators, providing transportation and sharing information that may not be otherwise accessible to abortion seekers.

The aim of this study was to explore abortion health professional and administrative staff perspectives regarding integration of doula support into abortion care services.

## Methods

### Study design

This is a qualitative study. We intended to do individual interviews with health care providers, however we pivoted to allow for focus groups when potential participants expressed a preference for that approach. The semi-structured guide included questions about participant location, clinical setting (hospital, freestanding clinic, family practice, etc.), profession, roles in abortion care, knowledge of and experiences with abortion doulas, understanding of the role, relevant workplace policies, perceived gaps in services, and potential doula contributions.

### Eligibility and recruitment

Eligible participants included abortion health professionals (physicians, nurses, social workers) or clinic administrative staff, regardless of prior experience with abortion doulas. We used purposive and snowball sampling. The Research Coordinator sent a recruitment email to potential participants identified via relevant professional organizations, reproductive healthcare clinics, regional abortion care networks, or specific individuals within our networks including the Contraception and Abortion Research Team (CART) [31]. Recipients could respond to schedule an interview or focus group at their convenience, in English or French. We encouraged participants to share the study information with peers. All eligible individuals that expressed interest within the data collection period were selected to participate. The research team sought to achieve diversity of professionals (physicians, nurses, social workers, clinic administrative staff) as well to obtain representation from across Canada in our recruitment. Prior experience with abortion doulas was not required.

### Data collection

We collected data between July 1 and October 1 2024. AM conducted interviews and focus groups, in-person or virtually via a secure video conferencing platform. A professional transcriptionist transcribed and anonymized the recordings. Interviews and focus groups were conducted until professional and geographic diversity of participants was obtained. Data collection stopped when the team determined from preliminary analysis that conceptual depth was achieved [32,33]. Participants had the option of an individual interview or focus group with members of their workplace team.

### Data analysis

The research team analyzed de-identified transcripts using a collaborative process of thematic analysis [34]. Three members reviewed raw data to familiarize themselves and identify emergent themes. Two team members independently conducted initial coding of a subset transcripts, to establish inter-coder agreement. Initial coding was done directly on transcript documents using comments and highlighting. The initial coding scheme was then adapted through discussion

with the team. After the second coding scheme was established collaboratively, one team member re-coded all data using the refined coding scheme. This iteration was then organized into a spreadsheet and distributed among the research team. Using the shared spreadsheet, three team members finalized key themes and sub-themes through discussion and synthesized findings. Throughout analysis, the research team did not identify any differences in themes emerging from focus groups versus individual interviews. Participants were not given the opportunity to review findings.

### Ethical considerations

The Research Ethics Board at the University of New Brunswick approved this study (approval number 2024−080). All participants received an informed consent form in advance and provided either written or verbal consent (documented by the researcher conducting the interview and reflected at the beginning of interviews in the audio recording).

## Results

We conducted five individual interviews and two focus groups, with a total of 18 participants, in four provinces across Canada: Nova Scotia, Ontario, British Columbia and Alberta. Participants included 10 registered nurses, two family physicians, one registered social worker, two executive directors, one unit aide, and two clinic clerks. The participants worked in many different types of environments, including: two dedicated abortion clinics in hospitals, a women's hospital, one independent family practice offering medication abortion, two free-standing abortion clinics and one organization that supports abortion-seekers and abortion care providers. Some participants worked in sites where support people, including doulas, were not permitted in the clinical space, although they could support patients in other ways; some had worked with doulas in clinic but policies changed to limit support people; and some had active formal or informal collaborations with abortion doulas. Some worked mostly in the context of first-trimester abortion, while others provided second and third trimester care. We did not collect demographic information due to the small size of this community and potential identifiability. Four participants were recruited via snowball sampling, and fourteen were recruited via purposive sampling.

We identified two key themes and five sub-themes (see Table 1).

### Theme 1: A Tricky Fit

With the exception of a few, most participants felt abortion doulas would be challenging to "fit" within abortion clinical roles, activities and spaces. Sub-themes included "Staff meet patient needs" and "Contextual constraints".

### Staff meet patient needs

Participants working at dedicated abortion clinics, both free-standing and in hospital perceived their existing teams to be very focused on and well-resourced to meet patient needs for emotional support and service navigation with clinical expertise. They believed adding a non-clinical person would not necessarily enhance services:

**Table 1. Themes and Sub-Themes.**

| Theme | Sub-theme | Illustrative quote |
|---|---|---|
| A Tricky Fit | Staff meet patient needs | *"I think we do a really good job of being that support for people while they're here if they need it."* [P6] |
| | Contextual constraints | *"It doesn't really fit with the clinical flow of our clinic."* [P5] |
| A Place for Abortion Doulas | Before and After and Outside | *"To offload any amount of burden somebody is experiencing in just trying to get healthcare"*. [P7] |
| | Tailored Support | *"We of course try to practice cultural humility/sensitivity but obviously aren't privy to all the nuances of different cultures and I think doulas could fit in there."* [FG1] |

*The nursing is important while the patients are here, they just do such a great job with patients for the counseling that as well as having the medical knowledge, as well as having the medical skills and managing in an emergency that all the shit they do that I can't, they're the best and a lay person cannot add to this. [FG1]*

Some participants felt the lack of standardized training, certifying bodies and regulation of the doula profession resulted in too much uncertainty about the support patients would receive. One participant knew a doula who received their training in the US, a very different political, clinical and legislative context. One said simply, *"not all doulas are created equally."* *[P3]*. For example, some participants worried doulas could unintentionally misinform abortion-seekers about pathways to access or about clinical matters:

*Wrong information is much more harmful than no information, so just making sure that [doulas are] equipped with the proper information…Abortion care is something that a lot of misinformation is shared about so I think that you know somebody that might be coming from the right place but is sharing wrong information can be more harmful than good. [FG1]*

Participant concerns regarding the lack of standardized training of abortion doulas contrasts those who expressed confidence in the training that clinic/hospital staff had received:

*I think just because of the training that the social workers have here, cause I mean and the nurses too, I mean most of them are very good at supporting the people […]I guess it would be beneficial to the individual if they developed a relationship with that person cause they don't know us so I understand that, but I think we do a really good job of being that support for people while they're here if they need it. [P6]*

While participants were aware of the benefits an abortion doula may bring to a patient, they perceived clinic and hospital staff as meeting patient support needs during their time in a clinical space.

Participants characterized the context of an abortion clinic as different from a labour and birth unit, where birth doulas often play a significant role as advocates for patient autonomy. Participants believed that the abortion care team was deeply committed to gender equity and freedom of choice:

*Here in the clinic it's such a small team and everybody is very highly trained, very much on the same page, whereas if you're at the hospital you don't know any random OB-GYN you may not know, it might be some old-school man [...] whereas like nothing like that is happening here, there's not the same need to have a patient be advocated for because everyone here does only this and we know, everyone here would hold anyone else accountable if they tried to do something weird. [FG1]*

## Contextual constraints

Not being a good "fit" in some cases was expressed as a concern with space and workflow, because of concerns integrating abortion doulas into the clinical environment might introduce delays, disrupt the flow of care, and impact safety:

*Often doulas have unrealistic expectations about the time that they would be able to spend with their patients in the clinic. Like…we want to serve as many patients as possible…We don't have a lot of space…we only have two procedure rooms and the patients are not spending very long at all in the procedure rooms. And the feedback that the nurses and our medical director gave me was that, like doulas are really wanting to spend a long time preparing the patients in*

*the procedure room, you know doing lots of relaxation, nothing bad, just it doesn't really fit with the clinical flow of our clinic. [P5]*

Many participants worked in clinics with policies limiting or prohibiting the presence of support people, to protect the privacy and physical safety of all patients and staff, which would also preclude the presence of doulas.

*Sometimes it feels like there is a tension between wanting people to feel supported and then also not wanting other people to feel triggered or like their privacy is being invaded [...] I don't know how to kind of hold everyone's needs in our support person policy. [P7]*

Participants believed it would be unfair to make an exception for one type of non-clinical support person (the abortion doula) and not others, such as a boyfriend, mother, or friend.

*If people are in the waiting room they see someone come in with a doula and someone says well why can't I bring my friend in and you say well they opted to have a doula with them, you know so because they paid this person to be with them, they get something that you don't get, that's another thing that I don't like. [P5]*

Participants who had worked in environments that allowed support people described challenges managing the dynamic between patients and their support people, who could express fear, judgment, or coercion:

*Having worked in clinics before where patients can bring somebody in, I've never worked with doulas, it's always been a family member or something and their vow of support is shaky right because they're going through their own emotional thing, they may also be nervous about blood, medical environments, all of those things. So often they're not a support. [FG1]*

Participants in both hospital and free-standing clinics felt the heavy regulation of their clinical spaces and workforces presented barriers to integrating doulas, "*Especially in a [health authority] which we know is so legally-bound and policy-bound and they have their own regulations [...] There's a lot of bureaucracy. [FG1]*" Some hospital-based participants who worked with abortion doulas explained their institutional administration had concerns about their doula partnership "*in terms of privacy, confidentiality, access to records*" [FG2].

### Theme 2: A place for abortion doulas

Many participants suggested abortion doulas might be more helpful for abortion-seekers in settings not dedicated to abortion, such as a general hospital obstetrics/gynecology unit, or in settings where funding was inadequate.

*The spaces that are solely abortion clinics and are funded adequately so that they have the time and energy to spend there, I think a lot of the actual support that people need during the procedure they receive from the staff who work there. In hospital settings that might be doing lots of other things who don't have a dedicated abortion clinic or program or some of the clinics in [province] that aren't adequately funded so there's much more high volume, there could potentially be some need there [P3]*

### Before or after or outside the clinic

Participants could see an abortion doula "*role being really important in the decision-making process and perhaps afterwards…but I think we do a really good job of being that support for people while they're here if they need it. [P6]*". One suggested doulas could offer an alternative to predatory conversations with crisis pregnancy centres workers, who discourage abortion:

*We have a lot of anti-choice organizations that masquerade as being pro-choice when they're not, so I think there's a huge gap for people who are wanting to just talk to someone about finding out that they're pregnant and they're not sure what they want to do, so someone who's non-biased. [P6]*

Some participants believed clinics should collaborate with abortion doulas to ensure correct information is available:

*There's a real opportunity for doulas if they are able to partner with clinics because they would have then a real intimate and clear understanding of what's happening and so they can help reassure somebody or explain before somebody even arrives. Sometimes on the day of the procedure it can be overwhelming so if they have that relationship in advance then they can really be clear on answering whatever questions are going to come up. [P3]*

Participants also saw a navigation role for abortion doulas before care, "*to offload any amount of burden somebody is experiencing in just trying to get healthcare*".[P7] One participant worked with doulas who helped by "*booking hotels and stuff*". [FG1] Another recalled an abortion doula who "*even stayed in a hotel with someone who was from out of town or out of province*". [FG2] One participant explained how an abortion doula helped a young person without a credit card: "*they actually can't check into a hotel in many cases so it was imperative for us to try to find a doula support in that community to help them with some of those really technical but much needed pieces*". [P3] Another unmet patient need was childcare, although, as one participant expressed, "*I don't know if doulas could help or what would be the solution*". [FG1]

Participants also identified roles after a patient clinical encounter, such as supporting patients in their homes as they self-administer medication abortion and experience what can be an intense and long experience. As one said, "*If you're taking the meds, I can't be there with you when you're doing it.*" [FG1] Several believed abortion doulas could accompany patients to their homes after procedural abortion, when they may be groggy from anesthetic, as well as driving them to and from appointments.

Participants also acknowledged patients might need more emotional support post-abortion than typically offered by their clinics.

*You know that feedback about abortion care is often that it feels that patients have to reach out if they want follow-up as opposed to us reaching out to them to be like, "Do you need follow-up?" And just from the imperfect hospital world I understand how it's not necessarily practical for us to call every patient afterwards and be like, "Do you need follow-up?", but I think that it is significant to feel like you have somebody who is checking in with you without you needing to initiate that so I feel there's a role in kind of the continuum of somebody's experience. [P7]*

**Tailored support**

Participants felt abortion doulas may be valuable for equity-seeking groups, including newcomers:

*We of course try to practice cultural humility/sensitivity but obviously aren't privy to all the nuances of different cultures and I think doulas could fit in there. Particularly with the newcomer clients we have who are often isolated from their families. [FG1]*

And youth:

*Demographically-speaking there seems to be a bit more of a more of a struggle for younger patients and so they maybe have not come to terms with the situation, they are very certain about their decision but they I think would maybe benefit from more supports and they're maybe not always willing to be involved in the process of seeking the counseling. [P4]*

Participants with experience with abortion doulas shared examples of support for people experiencing homelessness, substance use, or violence. As one explained, "*if they're dealing with like neglect or some level of abuse that having that neutral abortion doula for support would be very beneficial. To literally hold their hand for the appointment.*" [FG2]

Participants believed abortion doulas may have a role in later gestational care, particularly when induction is used as it is like the labour induction process. Another suggested doula support for abortion-seekers when they travel for care, usually for later care:

> We also get a lot of patients who are referred from remote Northern Ontario, often times Indigenous patients who might live on a reserve and often are very little resources [sic] or little support from the community to access this care, and when someone is having to travel often by air to get to the city, it's hard enough for them to get here, never mind trying to find someone to come with them, to stay with them for a few days downtown, so I think those people really would benefit from additional support. [FG 2]

Participants recognized that equity-deserving populations should have access to additional services. Due to "*the historical experiences of Indigenous people in healthcare settings*" [P5], one clinic offers every Indigenous patient the support of a trained and paid Indigenous person. Participants also referred to the availability of interpreter services, acknowledging interpreters are "*not really a support person,*" but rather are "*more for the clinic to ensure that we can do our job.*" [P5]

Another believed doulas could support patients seeking termination due to fetal anomaly. "*Those are the patients who are grieving a lot during their appointments and you can just tell that nobody has really spent time with them in that grief.*" [P7]

Participants acknowledged, "*A lot of our patients have a lot of social challenges, whether that's unsupportive family members or unsupportive partners, and having an abortion doula present takes away a lot of the stress out of the way.*" [FG2] The doula role was perceived as one of friendship, "*sometimes they're the only person in that individual's life that they've told about the abortion so the act as like, you know, a friend kind of idea, somebody to bounce thoughts and concerns and feelings off.*" [P3]. Having that friend aware could also be empowering, "*It makes a difference to have somebody who is a witness to a thing that you're experiencing even if that's just what it is, I think that that can be very powerful for people. [P7]*

## Discussion

The aim of this qualitative study was to understand potential pathways for integration of doulas with abortion care from the perspective of abortion clinic health professional and administrative staff. To our knowledge, this is the first study on this topic in Canada. While many participants felt that incorporating doulas formally into the abortion clinic space may not be possible or necessary, most saw value for abortion doula support outside of the clinic setting, before and after abortion, and for tailored support for patients experiencing systemic oppression and discrimination or with complex social needs.

Several studies have investigated provider perspectives on the integration of birth doulas into hospital labour and delivery units both in Canada [35,36], and internationally [12,37,38]. Having a continuous support person during labour is recognized and recommended as an evidence-based intervention to improve birth outcomes [39–41], but there is little research on patient outcomes with abortion doula support [24,42].

Studies about integration of abortion doulas into clinical services, largely US-based, found early and surmountable challenges including consistency, communication, and the busy clinical setting [27]. Concerns about lack of standardized training and certification did emerge in our study. Scholarship on the potential for standardization in the doula profession is limited. One study by Semadeni et al. found tension between standardization and the need for flexible, diverse, and community-based doula trainings [43]. But the dominant barrier identified by participants was the constraints of the context. Accompaniment is a key tenet of doula support, and in abortion clinics it may be

limited due to the unique patient safety, confidentiality, and security concerns with the highly stigmatized abortion service [44], and because procedural abortion in the first trimester is very brief in nature, usually taking less than 10 minutes. Allowance for support people in abortion care spaces varies but tends to be strict given the priority of patient autonomy. However, models such as the one described from a hospital that implemented Indigenous patient support workers may suggest a shift in the landscape of non-clinical support within clinical settings and understanding of their importance.

As we found in our qualitative study with abortion doulas [30], participants in this study valued doula support for abortion-seekers experiencing intersecting systemic discrimination and barriers to accessing health care, who lack social-familial support, or who are facing the complexity of later gestational care or termination for fetal anomaly. However, participants described doula activities such as driving patients to and from appointments, providing childcare, and staying with people in hotel rooms, for which there may be safety and liability concerns. The reliance on uninsured and inconsistently trained volunteers to fulfill these needs is concerning and highlights a gap in wrap-around support for people accessing abortion.

Participants in this study perceived abortion care professionals to be focused on and well-resourced to meet patient needs for emotional support and service navigation, roles that doulas may fulfill in contexts without adequate public funding and dedicated personnel. Notably, this contrasts with findings from our qualitative research with abortion doulas [30], who described themselves as meeting unmet patient needs for support and navigation. US-based studies have identified patient-perceived gaps in abortion services related to unclear information, lack of decision-making support, and inadequate emotional support [45–47]. Perhaps, clinicians' emphasis on clinical concerns such as space and workflow overshadow the broader sociocultural dynamics that abortion doulas themselves are focused on including patient-centred care [27]. Contemporary Canadian research is needed to understand the experience of abortion-seekers in this country's unique decriminalized, universal, and publicly funded model of care, particularly as medication abortion continues to expand in use.

### Strengths and limitations

The strengths of this study include the professional diversity among participants and their distribution across Canada, providing varied perspectives from members of the abortion workforce in various clinical settings. We did not collect demographic data (e.g., age, years of experience, cultural background) due to potential identifiability, however we recognize this information could help contextualize participant perspectives and potential biases. The study is limited in that it does not include patient perspectives or those of workers in frontline social services. Both groups would provide insight into the experience of doula support in the context of abortion services. Patient perspectives are of particular importance to clarify the discrepancy between our research with abortion doulas [30], stating that they fill gaps to meet patient needs, and this study, in which health care providers perceive clinic and hospital staff as meeting patient needs. Directions for future research could include patient perspectives on seeking abortion with and without an abortion doula, as well as further exploration of success and challenges among clinic-doula partnerships.

### Conclusion

Our findings suggest abortion health professionals and clinic administrative staff perceive roles for doulas external to the clinical setting, such as through non-judgmental conversations as patients make decisions about abortion, and after receiving care, for support at home. They have concerns about the presence of doulas in the clinic due to policies against support people and because of the unregulated, unstandardized nature of doula practice. They articulated a role for doulas to provide support for equity-deserving groups.

## Author contributions

**Conceptualization:** Martha Paynter.

**Formal analysis:** Martha Paynter, Anja McLeod, Clare Heggie.

**Funding acquisition:** Martha Paynter, Wendy V. Norman.

**Investigation:** Anja McLeod.

**Methodology:** Martha Paynter, Clare Heggie.

**Project administration:** Anja McLeod.

**Supervision:** Martha Paynter, Wendy V. Norman.

**Writing – original draft:** Martha Paynter, Anja McLeod, Clare Heggie.

**Writing – review & editing:** Martha Paynter, Anja McLeod, Clare Heggie, Wendy V. Norman.

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
