## [Decision Letter · Decision Letter 0]

27 Jun 2025

Dear Dr. Paynter,

Thank you for submitting your manuscript to PLOS ONE. After careful consideration, we feel that it has merit but does not fully meet PLOS ONE’s publication criteria as it currently stands. Therefore, we invite you to submit a revised version of the manuscript that addresses the points raised during the review process.

We look forward to receiving your revised manuscript.

Kind regards,

Stephen R. Milford

Academic Editor

PLOS ONE

3. We note you have included a table to which you do not refer in the text of your manuscript. Please ensure that you refer to Table 1 in your text; if accepted, production will need this reference to link the reader to the Table.

Additional Editor Comments (if provided):

Reviewers' comments:

Reviewer's Responses to Questions

**Comments to the Author**

1. Is the manuscript technically sound, and do the data support the conclusions?

Reviewer #1: Partly

Reviewer #2: Yes

2. Has the statistical analysis been performed appropriately and rigorously?

Reviewer #1: Yes

Reviewer #2: Yes

3. Have the authors made all data underlying the findings in their manuscript fully available?

Reviewer #1: Yes

Reviewer #2: No

4. Is the manuscript presented in an intelligible fashion and written in standard English?

Reviewer #1: Yes

Reviewer #2: Yes

Reviewer #1: Dear Authors,

Thank you for submitting your manuscript, "A Qualitative Study of Abortion Professionals’ Perspectives on Doula Support," for review. Your work addresses an important and underexplored topic in abortion care, particularly within the Canadian context. Below, I outline several key points for your consideration to strengthen the manuscript further:

Clarity in Sampling and Participant Demographics While the use of purposive and snowball sampling is noted, additional details about the selection criteria (e.g., specific roles, prior experience with doulas) would enhance transparency.

Including demographic data (e.g., age, years of experience, cultural background) for participants could help contextualize their perspectives and potential biases.

Methodology and Data Collection The shift from individual interviews to focus groups is understandable but warrants further discussion. For example:

Were there differences in themes emerging from focus groups versus individual interviews?

How were power dynamics or dominant voices managed in focus groups, given the sensitive topic?

Clarify whether data saturation was achieved and how this was determined.

Thematic Analysis Process Provide more detail about the coding process: How were initial codes developed? Was inter-coder reliability assessed? Were any software tools used (e.g., NVivo)?

Consider including a table or figure summarizing themes/subthemes with illustrative quotes for clarity.

Address whether participants were given the opportunity to review transcripts or findings (member checking).

Balancing Perspectives The manuscript highlights challenges in integrating doulas but could better acknowledge positive experiences or successful models (e.g., the Indigenous support program mentioned briefly). Are there lessons from these examples?

While the use of thematic analysis, as described by Braun and Clarke (Reference 30), is well-justified and widely accepted, the description of the collaborative coding process lacks sufficient detail to fully evaluate its rigor. Specifically:

Inter-Coder Reliability and Consistency: It is noted that two members conducted initial coding and developed a coding scheme, but the manuscript does not clarify whether these members:

Independently coded a subset of transcripts to establish inter-coder agreement (e.g., via Cohen’s kappa or percentage agreement). Collaboratively coded from the outset (and if so, how discrepancies were resolved).

Clarifying this process would strengthen the reproducibility of the findings. Refinement of the Coding Scheme: The manuscript mentions that the team adapted the coding scheme through discussion, but it would be helpful to specify: How many iterations of refinement occurred. Whether the final coding scheme was applied systematically by a single coder or multiple coders.

Limitations and Future Research The exclusion of patient voices is noted as a limitation, but its impact on the findings could be discussed further. For instance, how might patient perspectives alter the interpretation of "staff meeting patient needs"?

Suggest concrete future research directions, such as: Studies comparing clinics with/without doula integration. Patient experiences with doula support in Canada.

Structural and Editorial Refinements Some direct quotes lack sufficient interpretation. For example, the quote about "wrong information" (p. 15) could be tied more explicitly to concerns about doula training standardization.

Ensure consistency in terminology (e.g., "equity-deserving groups" is used but not defined).

Reviewer #2: 1The manuscript clearly indicates that the study uses qualitative methods and centers on abortion professionals' views. The integration of abortion doulas into care models is an emerging issue, particularly in the Canadian context where no similar study exists

2.Appropriate use of thematic analysis with diverse Participants enhances validity. Participants include a mix of professionals (nurses, physicians, administrative staff), and come from varied provinces, which adds generalizability within Canada.The use of both focus groups and individual interviews enhances the depth and triangulation of perspectives. Thematic analysis is appropriate and clearly explained.

3.Primary qualitative data (sensitive interview transcripts) are not publicly deposited due to privacy protections.

4.The manuscript is generally well-written in standard academic English, but requires minor linguistic refinements to meet PLOS ONE's clarity standards. 

Additional comments:

1.Methods Transparency

Sampling: The shift from individual interviews to focus groups due to participant preference raises potential selection bias. Please:a) Specify how many participants were recruited via snowball vs. purposive sampling.b) Disclose any institutional affiliations (e.g., were all sites pro-choice organizations?)

2.Lack of Patient Perspective: The authors themselves acknowledge that patient voices are missing. Including patient experiences (or even those of doulas) would provide a more holistic view.

3.Over-Reliance on Clinic Settings: The emphasis on clinical concerns (space, workflow) may overshadow broader sociocultural dynamics or the ethical imperative for patient-centered support.

4.Training and Certification Discussion Could Be Expanded: Given that "lack of standardization" is a key concern, it would be helpful to briefly explore efforts to professionalize doula training.

The study makes an important contribution to reproductive health services research, particularly regarding equitable access for Indigenous and migrant populations highlighted in the results.

**Do you want your identity to be public for this peer review?** For information about this choice, including consent withdrawal, please see our Privacy Policy

Reviewer #1: **Yes: ** Marzieh Bagherinia

Reviewer #2: No

---

## [Author Response · Author response to Decision Letter 1]

14 Oct 2025

Dear Reviewers,

Thank you for your feedback and suggestions to improve the manuscript, “A Qualitative Study of Abortion Professionals’ Perspectives on Doula Support.” Please find our point-by-point response attached.

---

## [Editor Report · Decision Letter 1]

23 Oct 2025

A Qualitative Study of Abortion Professionals’ Perspectives on Doula Support

PONE-D-25-24905R1

Dear Dr. Paynter,

We’re pleased to inform you that your manuscript has been judged scientifically suitable for publication and will be formally accepted for publication once it meets all outstanding technical requirements.

Kind regards,

Stephen R. Milford

Academic Editor

PLOS ONE
---

## [Editor Report · Acceptance letter]

PONE-D-25-24905R1

PLOS ONE

Dear Dr. Paynter,

I'm pleased to inform you that your manuscript has been deemed suitable for publication in PLOS ONE. Congratulations! Your manuscript is now being handed over to our production team.

Kind regards,

on behalf of

Dr. Stephen R. Milford

Academic Editor

PLOS ONE